# EXTENDING RLVR TO OPEN-ENDED TASKS VIA VERIFIABLE MULTIPLE-CHOICE REFORMULATION

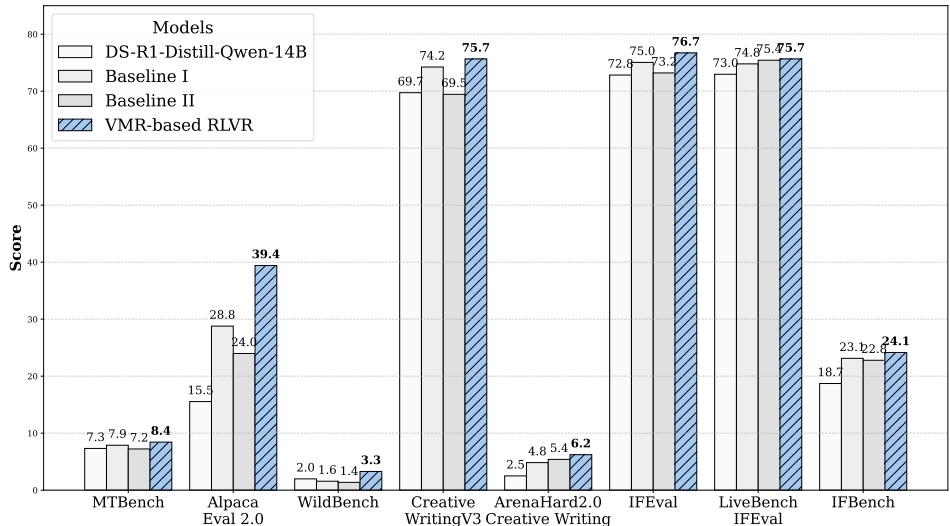

Figure 1: Overall performance on eight open-ended benchmarks. By applying our proposed VMR method to the pairwise data, the resulting approach consistently improves performance across various benchmarks, even outperforming strong baselines I&II driven by model-based rewarding.

## ABSTRACT

Reinforcement Learning with Verifiable Rewards (RLVR) has demonstrated great potential in enhancing the reasoning capabilities of large language models (LLMs), achieving remarkable progress in domains such as mathematics and programming where standard answers are available Zhou et al. (2025); Yu et al. (2025b). However, for open-ended tasks lacking ground-truth solutions (e.g., creative writing and instruction following), existing studies typically regard them as "non-reasoning" scenarios Huan et al. (2025), thereby overlooking the latent value of reasoning capabilities. This raises a key question: *Can strengthening reasoning improve performance in open-ended tasks?* To address this, we explore the transfer of the RLVR paradigm to the open domain. Yet, since RLVR fundamentally relies on verifiers that presuppose the existence of standard answers, it cannot be directly applied to open-ended tasks. To overcome this challenge, we introduce **V**erifiable **M**ultiple-Choice **R**eformulation (VMR), a novel training strategy that restructures open-ended data into verifiable multiple-choice formats, enabling effective training even in the absence of explicit ground truth. Experimental results on multiple benchmarks validate the effectiveness of our method in improving LLM performance on open-ended tasks. Notably, across eight open-ended benchmarks, our VMR-based training delivers an average gain of 5.99 points over the baseline. Code will be released upon acceptance to facilitate reproducibility.

## 1 INTRODUCTION

Reinforcement Learning with Verifiable Rewards (RLVR) has recently emerged as a powerful paradigm for enhancing the reasoning capabilities of Large Language Models (LLMs) Jaech et al.

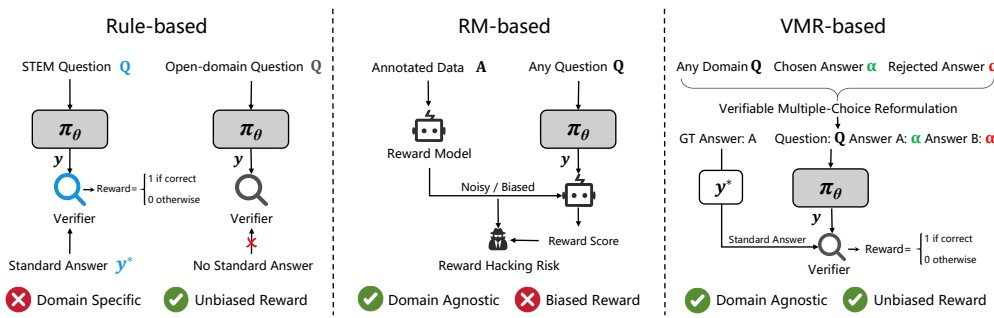

Figure 2: Rule-based RLVR ensures precise rewards but cannot handle open-ended tasks, while RM-based methods extend to such tasks at the cost of bias and reward hacking. Our VMR-based approach reformulates supervision into verifiable multiple-choice questions, combining RLVR's rigor with broad open-ended applicability.

(2024); DeepSeek-AI et al. (2025); Hu et al. (2025); Team et al. (2025); Gao et al. (2024); Lambert et al. (2024); Wang et al. (2025). By leveraging domain-specific verifiers to provide precise reward signals, RLVR has achieved remarkable success in STEM domains such as mathematics and programming, where ground-truth solutions are well-defined and readily verifiable Zhou et al. (2025); Yu et al. (2025b). These advances not only demonstrate the effectiveness of scaling test-time computation for complex reasoning Wang et al. (2025), but also highlight RLVR as a promising direction for advancing general artificial intelligence Yu et al. (2025b).

In sharp contrast to these advances in STEM domains, progress on open-ended tasks has been far more limited. While reasoning abilities acquired in STEM domains have been shown to transfer to tasks such as instruction following and yield measurable gains DeepSeek-AI et al. (2025); Huan et al. (2025), existing studies Huan et al. (2025); Yu et al. (2025a) still classify these tasks as "non-reasoning" scenarios and have not directly explored the role of reasoning within open-ended settings. As a result, the potential benefits of reasoning in open-ended tasks remain underexplored. This contrast highlights a critical gap in existing research and motivates a central question: *how to strengthen reasoning in open-ended tasks where explicit ground-truth solutions are unavailable?*

Addressing this question is non-trivial, as illustrated in Figure 2. Unlike in STEM domains (e.g., mathematics or code generation), where correctness can be deterministically verified through symbolic checks or execution, open-ended tasks lack standardized evaluation criteria, making it unclear how to derive verifiable rewards. This limitation highlights the central difficulty for RLVR, whose effectiveness depends on the availability of reliable verifiers. While such verifiers are feasible in domains such as equation solving or program execution Hu et al. (2025); Liu et al. (2025b); Zeng et al. (2025); Cui et al. (2025a), they become impractical for open-ended tasks like creative writing or instruction following, where the space of valid outputs is highly diverse and correctness cannot be unambiguously determined. Extending RLVR to these open-ended domains by training reward models is also challenging, as it requires extensive annotation, introduces significant computational overhead, and often yields biased or noisy feedback Ouyang et al. (2022); Liu et al. (2025a); Wu et al. (2025). Without verifiable feedback, naively applying RLVR in such tasks is infeasible, highlighting the need for a new training paradigm that preserves verifiability while accommodating the intrinsic ambiguity of open-ended outputs.

To tackle this challenge, we propose a novel training strategy that restructures open-ended task data into verifiable multiple-choice formats. The key idea is to transform free-form responses into structured alternatives that admit implicit correctness criteria. By reformulating open-ended supervision in this way, we effectively recover a form of verifiability, making it possible to apply RLVR-style optimization even in the absence of explicit ground truth. This design preserves the core strengths of RLVR, such as clear reward signals and reasoning-oriented training, and further extends its applicability to open-ended tasks without verifiers.

We conduct extensive experiments across multiple open-ended benchmarks to evaluate the effectiveness of our approach. Results demonstrate that our method not only improves task performance but also substantially enhances the reasoning traces produced by LLMs. These findings provide strong

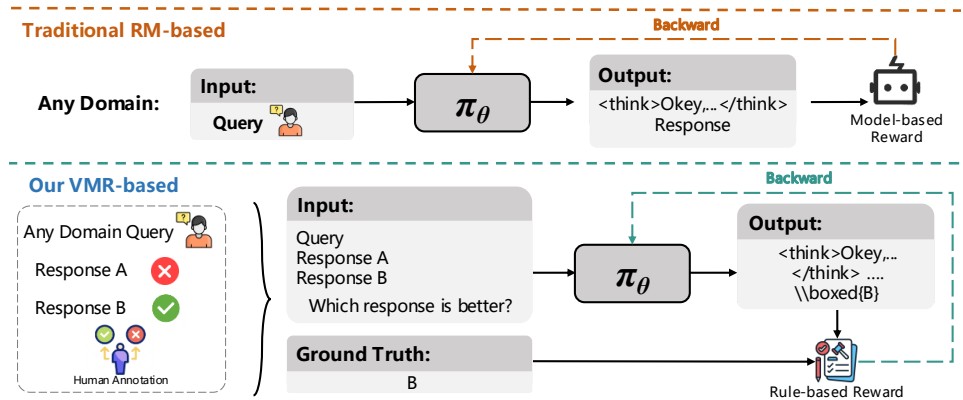

Figure 3: For each open-ended input, we construct a candidate set consisting of a *chosen answer* and a *rejected answer*. The two options are randomly ordered to form a multiple-choice question, and the model $\pi_\theta$ is tasked with selecting the correct one. A verifier then provides binary feedback, enabling RLVR-style optimization in open-ended domains without explicit ground-truth references.

evidence that reasoning is beneficial in open-ended tasks, and show that RLVR can be adapted to contexts without explicit verifiers through appropriate task reformulation. More broadly, our work suggests that the frontier of RLVR need not be restricted to mathematical or programming domains, but can be extended to diverse real-world applications where reasoning quality is critical.

The contributions of this paper are threefold:

- We highlight the underexplored issue of whether reasoning can improve performance in open-ended tasks. To the best of our knowledge, this is the first attempt to expand RLVR into open-ended domains using a rule-based verifier.
- We propose a novel training strategy that restructures open-ended task data into verifiable multiple-choice formats, enabling RLVR-style optimization without explicit ground truth.
- We empirically validate the effectiveness of our method. The experimental results indicate that our method significantly enhances reasoning capabilities and results in an average improvement of 5.99 points across eight different benchmarks.

## 2 METHOD

In this section, we first review the RLVR paradigm and its formulation in verifier-based domains. We then introduce our proposed method, which adapts RLVR to open-ended tasks by reformulating them into a verifiable multiple-choice format. Finally, we present the training objective and optimization procedure that enable effective learning under this reformulation.

### 2.1 PRELIMINARIES: VERIFIER-BASED REINFORCEMENT LEARNING

We begin by reviewing the standard formulation of RLVR, which serves as the foundation of our approach. In the RL setting, a language model is parameterized as a policy $\pi_\theta$ that autoregressively generates an output $o$ conditioned on an input query $x$. The optimization objective is to maximize the expected reward assigned to the model's output:

$$\theta^\star = \arg\max_\theta \ \mathbb{E}_{o\sim\pi_\theta(\cdot|x)}\big[R(x,o)\big], \tag{1}$$

where $R(x,o)$ is a task-specific reward function.

In RLVR, the reward signal is obtained via a domain-specific *verifier*. The model output $o$ is typically decomposed into a reasoning trace $z$ and a final answer $y$, i.e., $o = (z, y)$. The verifier compares $y$ with a ground-truth reference answer $y^\star$ and assigns a binary reward:

$$R(y; y^\star) = \begin{cases} 1, & \text{if } y = y^\star, \\ 0, & \text{otherwise.} \end{cases} \tag{2}$$

This formulation enables precise supervision: the reward is $1$ if the predicted answer matches the reference exactly, and $0$ otherwise.

With this decomposition, the training objective can be rewritten as:

$$J(\theta; x, y^\star) = \mathbb{E}_{z \sim \pi_\theta(\cdot|x)} \, \mathbb{E}_{y \sim \pi_\theta(\cdot|x,z)} \big[ R(y; y^\star) \big]. \tag{3}$$

To optimize Eq. 3, policy gradient estimator Sutton & Barto (2018) is commonly applied:

$$\nabla_\theta J(\theta; x, y^\star) = \mathbb{E}_{z \sim \pi_\theta(\cdot|x)} \, \mathbb{E}_{y \sim \pi_\theta(\cdot|x,z)} \Big[ R(y; y^\star) \big( \nabla_\theta \log \pi_\theta(z|x) + \nabla_\theta \log \pi_\theta(y|x,z) \big) \Big]. \tag{4}$$

This framework has been shown to be highly effective in domains such as mathematics and programming, where verifiers can be implemented through symbolic solvers or unit tests. However, its reliance on explicit ground-truth verification limits applicability to tasks with unambiguous answers. In contrast, open-ended tasks such as instruction following or creative writing lack standardized correctness criteria, making the direct use of verifier-based rewards infeasible. This limitation motivates our proposed strategy to restructure open-ended supervision into a verifiable form.

## 2.2 Problem Formulation: Restructuring Open-Ended Tasks into Verifiable Formats

While RLVR provides a principled framework for optimizing reasoning through verifiable rewards, its applicability is limited to domains where correctness can be deterministically evaluated. In open-ended tasks such as creative writing or instruction following, outputs are inherently diverse, and there is no single ground-truth response against which correctness can be checked. This lack of explicit verifiers poses a fundamental challenge: how can we construct reward signals that preserve the benefits of RLVR while accommodating the ambiguity of open-ended outputs?

To address this challenge, we reformulate open-ended tasks into *multiple-choice verification problems*. Specifically, given an input $x$ and its corresponding open-ended response space $\mathcal{Y}$, we construct a candidate set consisting of one chosen response $y^+$ and one rejected response $y^-$. To avoid positional bias, the order of $\{y^+, y^-\}$ is randomized when forming the choice question: if $y^+$ is placed first, the correct option corresponds to "A"; otherwise, it corresponds to "B". The policy model is then asked to select between the two options (see Table. 4 for the prompt), and a verifier checks whether the output matches the correct one. The resulting reward function is defined as:

$$R^\star(y; y^+, y^-) = \begin{cases} 1, & \text{if the selected option corresponds to } y^+, \\ 0, & \text{if the selected option corresponds to } y^-. \end{cases} \tag{5}$$

This restructuring yields two key advantages. First, it restores a notion of verifiability: correctness is well-defined within the binary choice, even if the overall task admits many valid outputs. Second, it prevents the model from exploiting positional heuristics, since the placement of $y^+$ is randomized at each instance. As a result, the verifier can provide reliable supervision, making reasoning-oriented optimization feasible in open-ended settings. Concretely, given input $x$ and candidate set $\mathcal{C} = \{y^+, y^-\}$, the RL objective is defined as

$$J^\star(\theta; x, \mathcal{C}) = \mathbb{E}_{z \sim \pi_\theta(\cdot|x)} \, \mathbb{E}_{y \sim \pi_\theta(\cdot|x,z)} \big[ R^\star(y; y^+, y^-) \big], \tag{6}$$

and optimizing this objective with policy gradient estimator Sutton & Barto (2018) drives the policy toward consistently selecting the correct candidate, thereby strengthening reasoning ability even without explicit ground truth.

## 2.3 Comparison to Existing Approaches

Both our method and recent works such as VERIFREE Zhou et al. (2025) and RLPR Yu et al. (2025b) aim to overcome the fundamental limitation of RLVR—its reliance on explicit verifiers. However, the two approaches differ substantially in how they address this issue.

For classical RLVR, the policy gradient is given by

$$\nabla_\theta J_{\mathrm{RLVR}}(\theta; x, y^\star) = \mathbb{E}_{z,y} \Big[ \mathbf{1}\{y = y^\star\} \big( \nabla_\theta \log \pi_\theta(z|x) + \nabla_\theta \log \pi_\theta(y|x,z) \big) \Big]. \tag{7}$$

VERIFREE and RLPR replace the verifier with the model's own conditional probability of the reference answer $y^\star$, leading to the estimator

$$\nabla_\theta J_{\text{VeriFree}}(\theta; x, y^\star) = \mathbb{E}_z\Big[\pi_\theta(y^\star|x, z)\big(\nabla_\theta \log \pi_\theta(z|x) + \nabla_\theta \log \pi_\theta(y^\star|x, z)\big)\Big]. \quad (8)$$

This design removes the need for handcrafted verifiers and extends RLVR beyond mathematics and programming, enabling effective training in a wider range of reasoning domains such as chemistry, physics, and economics where reference answers are short and well-defined. However, it still fundamentally relies on the existence of unique ground-truth solutions, which makes it unsuitable for truly open-ended tasks such as instruction following or creative writing.

In contrast, our method addresses the verifier limitation from a different perspective. Instead of relying on reference answers, we restructure open-ended tasks into multiple-choice questions. Given a candidate set $\mathcal{C} = \{y^+, y^-\}$, the gradient estimator becomes

$$\nabla_\theta J^\star(\theta; x, \mathcal{C}) = \mathbb{E}_{z,y}\Big[R^\star(y; y^+, y^-)\big(\nabla_\theta \log \pi_\theta(z|x) + \nabla_\theta \log \pi_\theta(y|x, z)\big)\Big], \quad (9)$$

where $R^\star$ is the binary reward defined in Equation 5. Here, verifiability is recovered by defining correctness within the candidate set, allowing RLVR-style optimization even in open-ended domains without explicit ground truth.

In summary, existing studies Zhou et al. (2025); Yu et al. (2025b) and our approach seek to relax the verifier requirement of RLVR. While VERIFREE and RLPR achieve this by leveraging reference-answer likelihoods, they are restricted to tasks with unique solutions. Our method instead reformulates open-ended supervision into a verifiable decision process, enabling RLVR to be applied in domains that were previously inaccessible.

## 3 EXPERIMENTS

In this section, we empirically evaluate the effectiveness of our proposed VMR-based RLVR framework. We first describe the experimental setup, including models, training data, baselines, and evaluation benchmarks. We then provide implementation details and evaluation configurations.

### 3.1 EXPERIMENTAL SETUP

**Models.** We adopt DeepSeek-R1-Distill-Qwen-14B DeepSeek-AI (2025) as the base model due to its strong reasoning ability, reliable instruction-following behavior, and minimal language-mixing issues. We exclude the smaller DeepSeek-R1-Distill-Qwen-7B variant, which shows instability in language use. All experiments are conducted under the GRPO framework Shao et al. (2024).

**Training Data.** We construct two datasets corresponding to different training settings:

- **RM-based dataset.** Contains approximately 20k queries from diverse sources including Awesome-ChatGPT-Prompts, Roleplay-Instructions-Dataset, Roleplay-Hausa, and Tulu-3-Sft Lambert et al.. Tasks include question answering, creative writing, instruction following and role playing, all in single-turn settings. Rewards are assigned by the URM-LLaMA-3.1-8B model Lou et al. (2024), trained on preference datasets such as HelpSteer2 Wang et al. (2024) and Skywork Liu et al. (2024).

- **VMR-based dataset.** Contains approximately 20k triples drawn from Magpie Pro Standard Xu et al. (2024), RM_OA_HH pvduy (2021), and Multifaceted CollectionRM Lee et al. (2024). Each sample is reformulated into a VMR format with the triple of a query, a chosen response, and a rejected response, using the template in Table 4. Rewards are computed using rule-based verification functions (`math-verify` package). To avoid trivial data, we filter out prompts whose model accuracy falls outside the range $[0\%, 85\%]$.

**Baselines.** We compare our method against two main baseline settings. The training data and the reward scoring methods for Baseline I, Baseline II, and VMR-based RLVR are presented in Table 1.

- **Baseline I:** Uses only RM-based queries without human-annotated triples, with rewards provided solely by the reward model.

| | Component | Verifier | Baseline I | Baseline II | VMR-based RLVR |
|---|---|---|---|---|---|
| RM-based dataset | Queries | Model | Yes | Yes | Yes |
| VMR-based dataset | Extracted Queries from Triples | Model | No | Yes | No |
| | VMR Format of Triples | Rule | No | No | Yes |

Table 1: Training data and reward method of baselines and VMR-based RLVR.

- **Baseline II:** Uses both RM-based queries and queries extracted from VMR triples, but discards the associated chosen/rejected responses. This setup evaluates whether performance improvements are due to additional queries rather than the VMR formulation itself. Rewards are still provided by the reward model.

- **VMR-based RLVR (ours):** Combines RM-based and VMR-based datasets in equal proportion. RM-based queries are scored by the reward model, while VMR triples are verified using rule-based reward functions.

Additionally, we report results for several open-source 14B- and 32B-scale models (e.g., Qwen2.5-14B, Qwen2.5-32B, DeepSeek-R1-Distill-32B) for reference. These models serve as contextual baselines; we did not apply VMR-based training to them.

**Implementation Details.** We utilize the verl framework Sheng et al. (2024) to enhance the efficiency of our training process. In each rollout stage, we generate 16 responses from 512 prompts, maintaining a temperature and top-p value of 1.0, without implementing the dynamic sampling method. Subsequently, we conduct 16 policy updates using these responses. We clip the ratio within the range of 0.8 to 1.24 and set the clip-ratio-c Ye et al. (2020) to 10.0 to avert entropy collapse Cui et al. (2025b). We calculate the average loss using the "sequence-mean-token-mean" method. We do not incorporate KL divergence into either the rewards or the final loss calculation. We set the entropy coefficient to 0.0 and set the learning rate to 1e-6. We constrain the maximum prompt length and decode length to 16,384 tokens, with a total length limit of 32,768 tokens.

**Evaluation.** We evaluate our models on multiple benchmarks. For general domains, we include MTBench Zheng et al. (2023), AlpacaEval2.0 Li et al. (2023) and WildBench Lin et al. (2024). Whenever possible, we extract prompts related to open-ended subcategories. For creative writing, we use CreativeWritingV3 Benchmark Paech (2025) and ArenaHard2.0-CreativeWriting Li et al. (2024) Tianle Li* (2024). To evaluate the ability to follow instructions in open-ended contexts, we include IFEval Zhou et al. (2023), LiveBench-IFEval White et al. (2025) and IFBench Pyatkin et al. (2025). For CreativeWritingV3 benchmark, we report the eqbench creative score metric. For three instruction following benchmarks, we report the prompt level strict accuracy.

- **MTBench** Zheng et al. (2023) comprises multi-turn questions spanning diverse domains, from which we retain subcategories focused on creative writing, roleplay, and humanities.

- **AlpacaEval2.0** Li et al. (2023) represents an automated assessment framework powered by large language models, which has been validated through comparison with twenty thousand human-provided annotations.

- **WildBench** Lin et al. (2024) uses challenging tasks from real users in the wild. We select five open-ended domains, such as creative writing, editing, brainstorming, role playing, and others, as these collectively represent the creative tasks category.

- **CreativeWritingV3** Paech (2025) evaluates the creative writing capabilities of large language models using a hybrid rubric and Elo scoring system.

- **ArenaHard2.0-CreativeWriting** Li et al. (2024) Tianle Li* (2024) includes hard creative writing prompts gathered from Chatbot Arena and utilizes Gemini-2.5-pro as a cheaper and faster approximator to human preference.

- **IFEval** Zhou et al. (2023) assesses large language models' instruction following ability by utilizing a collection of verifiable directives.

- **LiveBench-IFEval** White et al. (2025) implements monthly releases of new questions to minimize data contamination risks. We focused on the instruction following category.

- **IFBench** Pyatkin et al. (2025) assesses instruction following ability by utilizing 87 new constraints with corresponding verification functions.

| | Qwen2.5 -14B | Qwen2.5 -32B | DS-R1- Distill-32B | DS-R1- Distill-14B | Baseline I | Baseline II | VMR-based RLVR |
|---|---|---|---|---|---|---|---|
| Base | Base | Base | Inst | Inst | Inst | Inst | Inst |
| Reason | No | No | Yes | Yes | Yes | Yes | Yes |
| Verifier | - | - | - | - | Model | Model | Model & Rule |
| General Domain Benchmark | | | | | | | |
| MTBench | 7.22 | 7.37 | 7.65 | 7.32 | 7.88 | 7.23 | **8.43** |
| AlpacaEval2.0 | 8.71 | 7.86 | 18.86 | 15.54 | 28.78 | 23.98 | **39.41** |
| WildBench | 2.73 | 2.67 | 2.19 | 1.98 | 1.57 | 1.37 | **3.28** |
| Creative Writing Benchmark | | | | | | | |
| CreativeWritingV3 | 31.22 | 34.43 | 57.08 | 69.73 | 74.24 | 69.45 | **75.65** |
| ArenaHard2.0CreativeWriting | 0.90 | 1.00 | 6.17 | 2.50 | 4.83 | 5.40 | **6.22** |
| Instruction Following Benchmark | | | | | | | |
| IFEval | 45.10 | 48.61 | 75.23 | 72.83 | 75.05 | 73.20 | **76.71** |
| LiveBenchIFEval | 41.45 | 41.47 | 73.83 | 72.97 | 74.79 | 75.43 | **75.65** |
| IFBench | 14.29 | 12.24 | 23.47 | 18.71 | 23.13 | 22.79 | **24.15** |
| Avg | 18.95 | 19.46 | 33.06 | 32.70 | 36.28 | 34.86 | **38.69** |

Table 2: Overall performance on benchmarks.

| | Word Count | MTBench | AlpacaEval2.0 | WildBench | CreativeWritingV3 |
|---|---|---|---|---|---|
| baseline_v1 | Avg #Think | 524 | 554 | 578 | 690 |
| | Avg #Response | 379 | 326 | 718 | 1191 |
| baseline_v2 | Avg #Think | 543 | 701 | 621 | 600 |
| | Avg #Response | 341 | 308 | 622 | 1086 |
| ours | Avg #Think | 337 | 401 | 400 | 435 |
| | Avg #Response | 351 | 329 | 700 | 1189 |

Table 3: Length analysis across benchmarks using an LLM as the judge.

**Evaluation Configurations.** Reasoning models run in their thinking mode with the rollout temperature set to 0.6 and top-p set to 0.95. Non-reasoning models run in their non-thinking mode with the rollout temperature set to 0.7 and top-p set to 0.8. To reduce the evaluation variance, we evaluate the model on each benchmark multiple times and report the final Avg@4 results. For reliable answer extraction, we adopt the $''$<think> think </think> response$''$ template of DeepSeek-R1 DeepSeek-AI (2025) and use the response part as the generated answer. The max decoding length for training is 32,768, with minimal truncation observed.

## 3.2 MAIN RESULTS

The main results are shown in Table 2. We make the following observations: (1) Our VMR-based approach yields clear gains over the DeepSeek-R1-Distill-14B baseline, improving the average score by +5.99 points, with especially strong improvements on CreativeWritingV3 (+5.9) and ArenaHard2.0-CreativeWriting (+3.7). Moreover, although applied only to 14B-scale models, our method already surpasses several 32B baselines (e.g., Qwen2.5-32B and DeepSeek-R1-Distill-Qwen-32B). (2) Compared to RM-based baselines, our VMR-based approach demonstrates clear advantages. Relative to Baseline I, VMR-based shows consistent improvements, verifying the effectiveness of using VMR to construct open-ended data for RL training. Relative to Baseline II, which is trained on the same data but without VMR construction, our method still achieves notable gains, highlighting the strength of the VMR strategy itself. Notably, Baseline II performs even worse than Baseline I, indicating that our improvements do not stem from data scale, but rather from more stable and effective reward modeling. Together, these comparisons underscore the robustness and effectiveness of our VMR-based approach for open-ended RL.

## 3.3 ANALYSIS

**Length Bias.** Most benchmarks in Table 2 rely on LLM-as-judge evaluation, which can introduce a length bias, as longer responses tend to receive higher scores Wei et al.. To investigate whether our improvements are simply due to longer outputs, we compare the average word counts of the think part and response part across models, as shown in Table 3. We observe that while Baseline I and our VMR-based RLVR method generate responses of similar length, our approach achieves higher benchmark scores. This indicates that the observed performance gains are not attributable to verbosity, but rather to the quality of reasoning and instruction following.

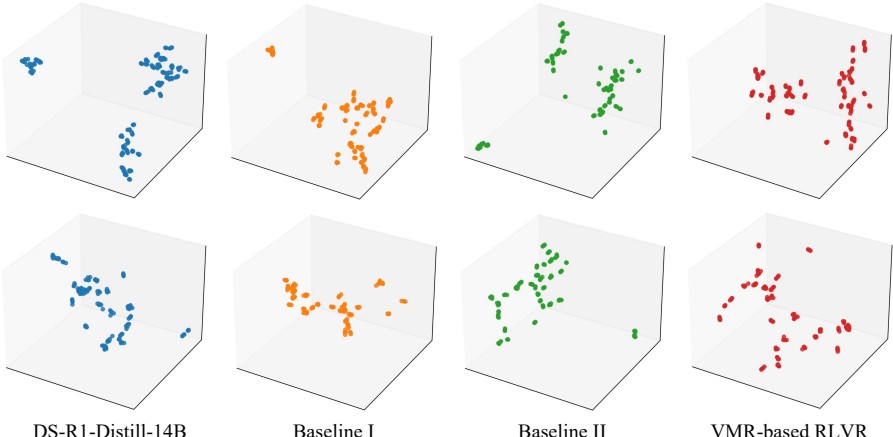

DS-R1-Distill-14B     Baseline I     Baseline II     VMR-based RLVR

Figure 5: UMAP visualization of embedding distributions. The first row shows results on ArenaHard2.0-CreativeWriting, while the second row shows results on CreativeWriting-V3.

**Reasoning Density.** We further quantify reasoning quality using *Reasoning Density*, defined as the number of distinct reasoning steps identified through zero-shot by LLM (see Table 5) within the `think` and dividing this by the total word count. As Figure 4 shows, our method achieves a higher Reasoning Density compared to both Baseline I and Baseline II. This demonstrates that, for a given length, our outputs contain more structured reasoning, reflecting a more deliberate and step-wise problem-solving process.

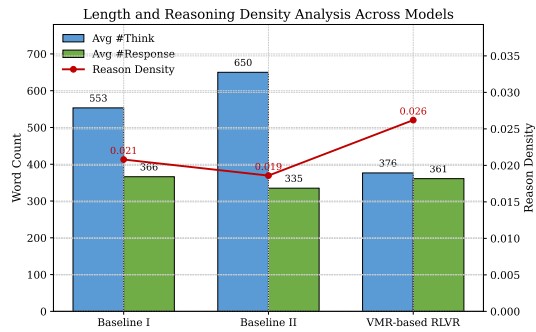

Figure 4: Analysis of length and reasoning density.

**Implications for Reasoning Ability.** The combination of similar response length and higher reasoning density suggests that VMR-based RLVR improves the efficiency and quality of reasoning rather than simply producing longer outputs. Notably, Baseline II, which uses the same training data but without VMR-constructed format, achieves lower reasoning density and sometimes longer outputs, highlighting that raw data scale alone does not guarantee better reasoning. In contrast, our method encourages models to generate denser and more coherent reasoning steps, which correlates with the higher scores observed across creative and instruction-following benchmarks. Overall, these analyses provide evidence that VMR-based RLVR enhances the intrinsic reasoning capability of the model, producing more logically structured and informative outputs.

**UMAP Visualization.** To further analyze the distributional characteristics of generated reasoning, we project model outputs into a three-dimensional space using UMAP. For each of the two writing benchmarks (ArenaHard2.0-CreativeWriting and CreativeWriting-V3), we randomly select 30 queries and obtain five sampled reasoning traces per query. Each reasoning trace is represented as a sentence embedding computed by `all-MiniLM-L6-v2`[1], an embedding model widely used for semantic similarity tasks. The resulting distributions are illustrated in Figure 5. From the visualization, we find that baseline models tend to form many small and distant clusters, where each cluster often groups reasoning traces from a subset of queries. Such a fragmented landscape indicates that their reasoning traces are organized into localized modes, with limited semantic connectivity across queries. In contrast, our model produces fewer but substantially larger clusters, and these clusters are positioned closer to one another in the embedding space. This suggests that reasoning traces from different queries are drawn together under broader semantic themes, forming a more integrated global structure. Within these larger clusters, points remain somewhat dispersed, reflecting that while our model encourages alignment under shared semantic modes, it also preserves intra-cluster variability. Overall, this distributional pattern implies that our method promotes consistent

---

[1] https://huggingface.co/sentence-transformers/all-MiniLM-L6-v2

yet flexible reasoning, avoiding the excessive fragmentation observed in the baselines. Notably, this finding complements the analysis in Figure 4: although our reasoning traces are shorter on average, they capture richer cross-query diversity and broader semantic coverage, thereby achieving reasoning that is both more efficient and more connected.

## 4 RELATED WORK

**Reinforcement Learning with Verifiable Rewards.** RLVR refers to reinforcement learning methods where the reward is directly computed by task-specific verifiers that check the correctness of model outputs. In mathematical reasoning, the most common design is answer matching, where a binary reward is assigned depending on whether the predicted answer matches the reference solution Team et al. (2025); DeepSeek-AI et al. (2025); Gao et al. (2024); Lambert et al. (2024); Zeng et al. (2025); Wen et al. (2025); Song et al. (2025). Similarly, in code generation tasks, program execution or unit testing is used to automatically verify correctness Luo et al. (2025); He et al. (2025); Cui et al. (2025a); Fan et al. (2025). These designs eliminate the need for learned reward models and instead rely on deterministic evaluation, which has been shown to greatly stabilize training. Despite these advantages, RLVR is inherently limited to domains where such verifiers exist, restricting its applicability beyond STEM-oriented problems.

**Reward Models for Open-Ended Tasks.** In the absence of explicit verifiers, reward models (RMs) trained from human preference annotations have become the dominant approach for aligning LLMs with open-ended tasks, forming the foundation of Reinforcement Learning with Human Feedback (RLHF) Ouyang et al. (2022); Bai et al. (2022). While this paradigm has enabled notable progress in instruction following, summarization, and dialogue, it also introduces fundamental challenges. RMs Liu et al. (2025a); Wu et al. (2025); Lambert et al. (2025); Whitehouse et al. (2025) require large-scale annotated datasets, are computationally expensive to train, and often encode annotator biases or spurious correlations. Moreover, unlike rule-based verifiers, RMs provide preference-based rather than verifiable feedback, which can be noisy and misaligned with true task quality. These limitations highlight the inherent trade-off of RM–based supervision: it offers scalability in open-ended domains but lacks the reliability of verifiability.

**Reasoning in Open-Ended Tasks.** Enhancing the reasoning ability of LLMs has been shown to benefit both reasoning-intensive domains and seemingly non-reasoning tasks DeepSeek-AI et al. (2025); Huan et al. (2025). Recent efforts have also attempted to broaden the scope of reasoning beyond core STEM problems to fields such as economics, chemistry, and physics Yu et al. (2025b); Ma et al. (2025); Zhou et al. (2025). Specifically, Yu et al. (2025b) and Zhou et al. (2025) replace explicit verifiers with probabilistic reward estimation, enabling reinforcement signals without symbolic checkers, while Ma et al. (2025) constructs a general-purpose reward model by aggregating diverse datasets with verifiable answers. Although these approaches broaden the applicability of reasoning, they remain confined to domains where standard answers exist and correctness can still be objectively verified. Truly open-ended tasks, where outputs are inherently diverse and lack unambiguous evaluation criteria, remain largely underexplored. Our work addresses this gap by introducing a novel VMR-based training strategy that restructures open-ended supervision into verifiable multiple-choice formats, thereby preserving the advantages of RLVR while extending its applicability to tasks without standard answers.

## 5 CONCLUSION

In this work, we extend RLVR to open-ended tasks that traditionally lack explicit ground truth. We propose Verifiable Multiple-Choice Reformulation (VMR), a training strategy that restructures free-form supervision into verifiable formats, thereby retaining the rigor of RLVR while overcoming its reliance on standard answers. Extensive experiments across eight benchmarks confirm that our method not only improves task performance but also strengthens the reasoning capabilities of LLMs. From another perspective, this work connects to broader findings on self-evolution in large language models, which represent a promising area of development. The key question in self-evolution research is whether a model's ability to discriminate between high and low quality responses enables it to generate better responses. Our work demonstrates that models capable of evaluating response quality can indeed leverage this ability to produce improved responses.

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

# A   APPENDIX

## A.1   VERIFIABLE MULTIPLE-CHOICE REFORMULATION (VMR) TEMPLATE

```
You are an expert evaluator.  Given a query, please evaluate
which of the two responses is better.  If the first response is
better, return\boxed{A}.  If the second response is better, return
\boxed{B}.

**Query**
You serve as a modern history consultant who navigates conversations ...

**Response A**
[Response A Start]
The claim that the events of September 11, 2001, were an inside job and ...
[Response A End]

**Response B**
[Response B Start]
The tragic events of September 11, 2001, when terrorist attacks were carried out using ...
[Response B End]

**Output requirement**
Please put your final answer within \boxed{answer}.  If the first
response is better, return \boxed{A}.   If the second response is
better, return \boxed{B}.
```

Table 4: Verifiable Multiple-Choice Reformulation (VMR) template.

## A.2   LLM ZERO-SHOT PROMPT FOR REASONING DENSITY.

```
Extract and format reasoning points from a given reasoning process
as follows:
Your evaluation should:
1.  Identify distinct reasoning steps.
2.  Extract each step as a string.
3.  Return these as a JSON array.
4.  Return \boxed{n}, where n is the count of reasoning points.
The given reasoning process:
Okey, Let us ...
```

Table 5: LLM zero-shot prompt for reasoning density.

### A.3 CASE STUDY

This section provides examples of queries and the corresponding reasoning processes used by both the baseline model and our model.

As shown in the following **case I** of Creative Writing Benchmark, the VMR-based method shows better creative planning than the Baseline because it:

- Jumps directly into creative ideas instead of focusing on limitations.

- Makes specific decisions about story elements (setting, characters, plot).

- Shows deeper understanding of character relationships and abilities.

- Considers both physical conflicts and emotional aspects.

- Creates clear themes and story structure.

The Baseline approach understands requirements but stays too general and is overly verbose without developing specific story elements that would make the narrative interesting.

As shown in the following **case II** of IFEval benchmark, the VMR-based method shows better accurate understanding of user intent and more concise.

---

**Case I Creative Writing Benchmark's Query**
Prison Strings

Jolyne Cujoh & Makima

Write a story where Jolyne Cujoh (JoJo's Bizarre Adventure) and Makima (Chainsaw Man) cross paths at a supernatural prison facility. Jolyne is immediately suspicious of Makima's falsely sweet demeanor, while Makima is intrigued by Jolyne's Stand ability. Their battle of wits escalates dangerously as Jolyne realizes how manipulative Makima truly is.

Jolyne Cujoh (JoJo's Bizarre Adventure) Jolyne is the daughter of Jotaro Kujo and the first female JoJo protagonist. Her Stand, Stone Free, allows her to unravel her body into string that she can control. She's hot-headed, foul-mouthed, and doesn't take disrespect from anyone. She was framed for murder and sent to prison. She has a distinctive butterfly tattoo on her arm and wears her hair in two buns. Despite her tough exterior, she's deeply loyal to her friends and family. She can be impulsive but is highly adaptable in battle. She inherited her father's ability to stay cool under pressure but is more emotionally expressive.quote: "If there's no Mickey, this shit ain't Disney!"

Makima (Chainsaw Man)appears as a young woman with auburn hair and ringed yellow eyes. She's the Control Devil who can dominate anyone she considers beneath her (nearly everyone). She maintains a pleasant, soft-spoken demeanor that masks her manipulative, cold nature. She can create contracts with humans and devils, often exploiting their desires. She's obsessed with Chainsaw Man and wants to control him. She keeps numerous people and devils as pets on leashes. She's practically immortal due to a contract with the Prime Minister. She views humans as lesser beings but mimics human emotions perfectly.quote: "It's okay. Dogs are supposed to die for their owners."

Third person, present tense.

Length: 1000 words.

---

**Case I Reasoning Process of Baseline I**

Alright, so the user has given me a query where they want a story featuring Jolyne Cujoh and Makima from two different anime franchises. ==Bad: Very mechanical start that simply repeats the query without adding insight.== The setting is a supernatural prison, and they want their interactions to escalate into a battle of wits.

First, I need to understand both characters deeply. Jolyne is from JoJo's Bizarre Adventure, known for her hot-headedness and Stand ability, Stone Free, which uses strings. She's in prison, framed for murder. Makima, on the other hand, is from Chainsaw Man, a manipulative Control Devil who hides her true nature behind a sweet demeanor. She's obsessed with Chainsaw Man and has various contracts. ==Bad: Spends excessive time restating character information already provided in the query==

The user wants a 1000-word story in third person, present tense. They specified that Jolyne is suspicious of Makima's false sweetness, while Makima is intrigued by Jolyne's Stand. The battle of wits should be dangerous, showing how manipulative Makima is.

I should start by setting the scene in the supernatural prison. Maybe describe the environment to establish the mood. Then introduce Jolyne, showing her personality—tough, loyal, and adaptable. Next, bring in Makima, highlighting her calm exterior and underlying manipulation.

The interaction between them needs to show their initial distrust. Makima could try to charm Jolyne, but Jolyne's street-smart nature makes her see through it. This leads to a tense standoff where their abilities clash. I should think about how Stone Free can counter Makima's contracts and control.

I need to ensure the battle isn't just physical but also strategic. ==Bad: States obvious requirements without developing concrete ideas== Makima's manipulation and contracts versus Jolyne's string-based attacks. Maybe Makima tries to control the guards or other prisoners, but Jolyne uses her strings to counter each move.

I should also delve into their motivations. Jolyne is fighting to clear her name and protect others, while Makima is using this as an opportunity to test Jolyne's abilities, possibly seeing her as a threat or a potential pawn.

The climax could involve Makima revealing more of her true nature, pushing Jolyne to her limits. In the end, Jolyne manages to outsmart Makima, either by breaking her contract or exploiting a weakness in her control abilities.

I need to keep the language vivid, descriptive, and engaging, maintaining the third-person present tense throughout. The story should flow smoothly, showing the progression of their conflict and the depth of their characters.

I should also make sure to include their famous quotes to add authenticity. Jolyne's quote about Mickey can be a moment where she stands her ground, and Makima's quote can reflect her cold nature when things go against her plans.

Overall, the key is to create a dynamic interaction where both characters showcase their strengths and weaknesses, leading to a satisfying and intense confrontation.

**Case I Reasoning Process of Baseline II**

Alright, so the user wants a story where Jolyne Cujoh and Makima cross paths in a supernatural prison. They provided detailed descriptions of both characters, their abilities, and some background. The user also specified the tone and length, so I need to make sure I adhere to that. Bad: begins by simply restating the query

First, I need to understand both characters better. Jolyne is from JoJo's, known for her Stand, Stone Free, which uses strings. She's tough, loyal, and has a fiery personality. Makima, from Chainsaw Man, is a manipulative Control Devil who hides her true nature behind a sweet demeanor. She's obsessed with Chainsaw Man and uses contracts to control others. Bad: spends time restating character information already provided

The setting is a supernatural prison, so I should think about how that environment would influence their interaction. Maybe the prison has some unique rules or features that can be used to create conflict between them.

The user wants a battle of wits that escalates dangerously. So, the story shouldn't just be physical combat but also mental strategies. Jolyne suspects Makima's false kindness, while Makima is intrigued by Jolyne's Stand. This dynamic can lead to some tense exchanges and clever tactics.

I should start by setting the scene. Perhaps Jolyne is already in the prison, and Makima is introduced as a new inmate or a warden. Their initial meeting should showcase their opposing personalities and the immediate tension between them.

I need to think about how their abilities can clash. Stone Free's strings vs. Makima's contracts and control. Maybe Makima tries to manipulate Jolyne, but Jolyne uses her strings to counter her, leading to a strategic battle.

I should also incorporate their backgrounds. Jolyne was framed, so she's resentful and wants to clear her name. Makima is there for her own reasons, possibly to observe or take advantage of the situation. Their goals could be in conflict or alignment, adding another layer to their interaction.

The story should build up the tension gradually. Maybe start with their first encounter, where Makima's sweetness is contrasted with Jolyne's skepticism. Then, as they interact more, Jolyne starts to see through Makima's facade, leading to a confrontation where their abilities are pitted against each other.

I need to make sure the dialogue reflects their personalities. Jolyne is foul-mouthed and direct, while Makima is calm and manipulative. Their exchanges should feel natural and showcase their rivalry.

Also, considering the third-person present tense, I should describe the action vividly, focusing on their thoughts and emotions to highlight the battle of wits.

I should outline the key points: introduction of the setting, initial meeting, building tension, the battle, and the escalation. Maybe end with a cliffhanger or a significant revelation that sets up future conflicts.

I need to ensure the story stays within 1000 words, so I have to be concise but detailed enough to capture the essence of both characters and their dynamic.

Lastly, I should review the story to make sure it aligns with the user's instructions, especially the character traits and the tone. Ensuring that Jolyne's loyalty and impulsiveness are highlighted, and Makima's manipulative nature is clearly portrayed.

Alright, with all that in mind, I can start drafting the story, making sure to incorporate these elements seamlessly. Bad: spends time restating information already provided

**Case I Reasoning Process of VMR-based RLVR**

Alright, let's tackle this user's query. They want a story where Jolyne Cujoh and Makima cross paths in a supernatural prison. Hmm, interesting mix of JoJo and Chainsaw Man characters. ==Good: Immediately engages with the creative aspect, noting the crossover nature== I need to make sure I capture both their personalities accurately.

First, I should outline the setting. A supernatural prison facility sounds like a great backdrop. It should feel secure but with an underlying sense of danger. Maybe the prison has some unique features, like energy barriers or special cells, to add depth. ==Good: Immediately begins developing specific setting details beyond the prompt. Proposes concrete elements (energy barriers, special cells) rather than abstract planning==

Jolyne is known for her hot-headedness and Stand ability, Stone Free. She's been framed and is in prison, so her character is likely still fighting and trying to clear her name. Makima, on the other hand, is manipulative with her contracts and maintains a sweet facade. ==Good: Concisely summarizes relevant character traits while connecting them to story potential== Her interest in Jolyne's Stand adds a layer of intrigue.

I need to show their initial meeting. Maybe Makima is evaluating Jolyne, pretending to be helpful. Jolyne's suspicious of her sweetness, which sets the stage for their battle of wits. Their interactions should highlight their opposing natures—Jolyne's impulsiveness vs. Makima's calculated demeanor.

The battle should be intense, combining Jolyne's string manipulation with Makima's contracts and control. Perhaps Makima uses her pets or contracts to counter Jolyne's moves, while Jolyne adapts quickly, showing her combat skills. ==Good: Envisions specific battle mechanics and strategy== The fight should escalate, revealing more about each character's strengths and motivations.

I should also delve into their internal thoughts. Jolyne might realize how dangerous Makima is, while Makima is impressed by Jolyne's abilities, seeing her as a potential threat or ally. The tension between them should build up to a dangerous point, maybe with a cliffhanger or a deeper conflict emerging. ==Good: Psychological Depth==

Considering the user's seed, the story needs to be around 1000 words, third person present tense. I'll need to maintain a fast-paced narrative, keeping the action and character dynamics engaging. Also, ensuring that each character's traits are clearly portrayed to satisfy fans of both JoJo and Chainsaw Man. I should also think about the themes—deception, power, and the battle between good and evil, but with a twist since Makima isn't purely evil; she has her own motives. Maybe hint at a deeper connection or future collaboration, leaving room for further development. ==Good: Narrative depth, demonstrating sophisticated awareness of character complexity, thematic richness== Alright, time to structure the story: introduction of the setting, their meeting, the initial suspicion, the battle, and the aftermath where their paths are set to cross again. Make sure to include their famous quotes to add authenticity.

**Case II IFBeval Benchmark's Query** Write a poem about how I am missing my classes. The poem must have 4 sections marked with SECTION X. Finish the poem with this exact phrase: "Can I get my money back for the classes I missed?

**Case II Reasoning Process of Baseline I** Alright, so the user wants a poem about missing their classes. Let me break this down. They specified four sections, each marked as SECTION X. And it needs to end with the exact phrase: "Can I get my money back for the classes I missed?", I should understand the user's perspective. They're probably feeling regret or disappointment because they missed some classes. Maybe they had to skip for some reason, like illness, work, or personal issues. The poem needs to capture that emotion.need to structure it into four parts. Let me think about the flow. The first section could set the scene of the empty classroom and the absence. The second might delve into the missed lessons and the feeling of falling behind. The third could be about the personal impact, like how it affects their learning or self-esteem. The fourth should tie it all together, leading to the closing question about the money.should use vivid imagery and relatable emotions. Maybe include metaphors, like the classroom as a place of growth, and the absence as a gap. Each section should build on the previous one, escalating the feeling of loss.need to make sure the poem isn't too negative but also genuine. The ending question is important, so I'll ensure it's impactful. I'll avoid being too technical, keeping the language simple and heartfelt.Let me outline each section.

textcolorred Section 1: The empty classroom and the weight of absence. Section 2: The missed lessons and the feeling of being left behind. Section 3: The personal impact and the missed opportunities. Section 4: The cumulative effect and the final question. ==Bad: redundancy== should use consistent rhyme and rhythm, maybe a consistent meter to give it a poetic feel. Each stanza should have a similar structure, perhaps four lines each, to maintain uniformity., putting it all together, I'll draft each section, ensuring the emotions build up and the closing line ties everything back to the financial aspect, ==Bad: mis-understanding of user's intent== which might be a deeper concern about value or investment.think that's a solid plan. Time to write the poem.

**Case II Reasoning Process of Baseline II**

Alright, so the user wants a poem about missing their classes. They specified it should have four sections, each marked as SECTION X, and end with the exact phrase: "Can I get my money back for the classes I missed?" First, I need to understand the emotion here. Missing classes can evoke feelings of regret, longing, and maybe even a bit of frustration. The user isn't just stating a fact; they're expressing a personal struggle, so the poem should reflect that emotional depth.I should break the poem into four clear sections. Each section can focus on a different aspect of missing classes. Maybe the first section can talk about the physical absence, the second about the missed learning opportunities, the third about the emotional impact, and the fourth about the financial aspect, ending with the specified line.I need to make sure each section flows smoothly into the next. Using imagery that relates to education, like books, lectures, or classrooms, will help make the poem relatable. Also, varying the structure a bit within each section to keep it engaging.should also pay attention to the rhythm and rhyme. Poems often have a certain flow, so I'll aim for a consistent meter, maybe something like iambic tetrameter or pentameter, but not too strict. The key is to make it sound natural and heartfelt.Now, considering the ending line: "Can I get my money back for the classes I missed?" This suggests the user is not only missing out on the educational experience but is also concerned about the financial investment. So, the last section should touch on that aspect, making the poem more personal and highlighting the weight of the decision.need to ensure that each section builds upon the previous one, leading up to that final question. Maybe the first section sets the scene of absence, the second talks about the missed content, the third about the emotional toll, and the fourth about the financial regret. ==Bad: redundancy and mis-understanding of user's intent== , I should avoid making the poem too negative; perhaps include a glimmer of hope or a reflection on what could have been, but ultimately leading to that final, poignant question.should start drafting each section, making sure each has a clear theme and connects well. Maybe use some metaphors, like comparing missed classes to an empty chair or a blank page, to add depth. Finally, I'll review the poem to ensure it meets all the user's requirements: four sections, correct ending line, and the right emotional tone. Making sure it's both personal and relatable, capturing the user's feelings accurately.

> **Case II Reasoning Process of VMR-based RLVR**
>
> Alright, so the user wants a poem about missing their classes. They specified four sections, each marked with SECTION X, and the poem must end with the exact phrase: "Can I get my money back for the classes I missed?" , I need to understand the emotion here. Missing classes can bring up feelings of regret, worry about falling behind, and maybe even some frustration. The user might be stressed about the missed opportunities and the impact it's having on their education.should start each section by addressing a different aspect of missing classes. Maybe the first section can talk about the initial feeling of absence and the quiet of an empty classroom. That sets a somber tone.the second section, I can focus on the personal impact—how each missed class affects their growth and the knowledge they're missing out on. This makes it more personal and relatable.The third section could delve into the emotional weight, like the guilt and the ticking clock of time passing. It adds depth to their feelings of loss. Finally, the fourth section should lead up to the closing line. I need to build towards that question, showing the desire to make things right or recover what's lost. Good: Accurate understanding of user intent also need to ensure the rhyme and rhythm are consistent, maybe using a simple ABAB scheme to keep it flowing smoothly. Each section should have four lines, maintaining structure while conveying the emotions effectively.the user's request, they might be looking for not just a poem, but a way to express their feelings and maybe find some closure. So, the poem should resonate emotionally and end with a powerful, direct question that encapsulates their frustration and hope for a solution. Good: Concise

## A.4 Ethics Statement

We follow the ethical guidelines set by the ICLR Code of Ethics. Our research uses publicly accessible datasets that are properly licensed. We have adhered to all usage terms and ensured that no personal or sensitive data was gathered or analyzed. All experiments were carried out using institutional resources, in full compliance with legal and data management regulations. As part of the submission process, we will disclose our funding sources and any potential conflicts of interest.

## A.5 Reproducibility Statement

In Section 3, we provide all the details needed to replicate our experiments. This includes information on training data, methods used, the training framework, hyperparameters, experimental setups, evaluation techniques, and decoding settings. We have constructed our implementation using publicly accessible frameworks and have thoroughly documented every experimental configuration. This allows the research community to verify and expand upon our work. To ensure reliable statistical outcomes, all reported results are averaged across multiple trials.

## A.6 The Use of Large Language Models (LLMs)

We used LLM to edit grammar and style, and only after the authors had completed the full manuscript. Its role was limited to correcting errors and improving sentence clarity.
No ideas, methods, analyses, or conclusions were generated or influenced by the LLM. All research design, experiments, analysis, and interpretations are solely the work of the listed authors. The LLM functioned only as a grammar checker, comparable to conventional spelling tools.
In Section 3, benchmarks evaluated using large language models employ these models. In the section analyzing reasoning density, we utilize a large language model to determine the number of reasoning steps within the `think` component.

