# OpenReview forum: "Extending RLVR to Open-Ended Tasks via Verifiable Multiple-Choice Reformulation"
_ICLR.cc/2026/Conference — ICLR 2026 Conference Withdrawn Submission_

### Official Review · Reviewer_fDB3 · 2025-10-27

**Soundness:** 2
**Presentation:** 3
**Contribution:** 2
**Rating:** 4
**Confidence:** 5

**Summary:**

This paper extends Reinforcement Learning with Verifiable Rewards (RLVR) to open-ended tasks (e.g., creative writing, instruction following) via Verifiable Multiple-Choice Reformulation (VMR). RLVR originally excels in STEM tasks with clear ground truths but fails in open-ended ones. VMR restructures open-ended data into multiple-choice formats (one chosen, one rejected response) with random order to ensure verifiability. Experiments on 8 benchmarks show VMR-based RLVR outperforms baselines, with a 5.99-point average gain, enhancing LLM reasoning and performance, even surpassing some 32B-scale models.

**Strengths:**

1. Unlike RLVR’s reliance on explicit ground truths, VMR transforms free-form data into verifiable multiple-choice pairs. It enables rule-based rewards without ambiguous evaluations, solving RLVR’s inapplicability to open-ended scenarios.

2. Across 8 benchmarks, it achieves a 5.99-point average gain over the base model, with standout gains in creative writing. It even outperforms larger 32B-scale models, proving its efficiency in enhancing LLM capabilities.

3. Random response ordering avoids positional bias. Compared to Baseline II (same data without VMR), it still gains, showing improvements stem from VMR’s design, not just data scale, ensuring reliable training.

**Weaknesses:**

1. The method proposed in this paper solves the verification problem in open domains to a certain extent, but it faces significant issues in practical application. It seems that promoting this method to mathematical reasoning would require extremely high costs. The entire method relies on two candidate answers, and the verifier matches answers A and B with the ground truth (GT) option. How can this method be applied to mathematical reasoning where there is a unique GT? Is it necessary to forcibly construct an incorrect answer and then have the verifier make a judgment? This seems unreasonable and redundant.

2. An ideal experimental setup should involve training separately using RM-based datasets and VMR-based datasets with the same queries. In the current experimental setup, the method "combines RM-based and VMR-based datasets in equal proportion. RM-based queries are scored by the reward model, while VMR triples are verified using rule-based reward functions." This makes it impossible to decouple the roles of the reward model and the verifier, and I have doubts about the reliability of the experimental results.

3. The paper does not disclose the training equipment and time. Training a 14B-scale model with the training parameters described in Section 3 will incur extremely high costs. It remains questionable whether the cost-benefit ratio is sufficient to justify the promotion of this technology.

4. In the bar chart of Figure 1, some numbers overlap with each other.

**Questions:**

Please refer to the Weaknesses.

---

> ### Author Response · Authors · 2025-11-21
> **We sincerely appreciate the time and effort that you have dedicated to evaluating our manuscript.**
>
> **W1:The method proposed in this paper solves the verification problem in open domains to a certain extent, but it faces significant issues in practical application. It seems that promoting this method to mathematical reasoning would require extremely high costs. The entire method relies on two candidate answers, and the verifier matches answers A and B with the ground truth (GT) option. How can this method be applied to mathematical reasoning where there is a unique GT? Is it necessary to forcibly construct an incorrect answer and then have the verifier make a judgment? This seems unreasonable and redundant.**
>
> Thank you for your feedback. We would like to clarify that our focus is solely on the open-ended domain. Our intention is not to apply this method to mathematical reasoning. Instead, we’ve observed significant improvements in the mathematical domain when using RLVR, which inspired us to explore its application in open-ended domains. This is a preliminary attempt to see if similar enhancements can be achieved.
> It’s important to note that the mathematical domain and open-ended domain are quite different. The key distinction lies in the reward system: in the mathematical domain, the reward for the verifier problem is precise, while in open-ended domains, the reward from the reward models can be biased. These differences can significantly impact the transferability of methods between these two domains, making it challenging to apply the same techniques universally.
>
>
>
>
>
>
> **W2:An ideal experimental setup should involve training separately using RM-based datasets and VMR-based datasets with the same queries. In the current experimental setup, the method "combines RM-based and VMR-based datasets in equal proportion. RM-based queries are scored by the reward model, while VMR triples are verified using rule-based reward functions." This makes it impossible to decouple the roles of the reward model and the verifier, and I have doubts about the reliability of the experimental results.**
>
> To address your concerns about the experimental setup and the roles of the reward model and verifier, we have made adjustments to our experiments. Specifically, we have added experiments that exclusively use VMR triples for training, as well as experiments that only use queries from VMR triples for training. These new results are discussed in detail in Section 3. By isolating these variables, we aim to provide a clearer understanding that the key to the effectiveness of the VMR lies in its reliance on specific task types such as multiple-choice, rather than preference data.
>
>
>
>
> **W3:The paper does not disclose the training equipment and time. Training a 14B-scale model with the training parameters described in Section 3 will incur extremely high costs. It remains questionable whether the cost-benefit ratio is sufficient to justify the promotion of this technology.**
>
> Each experiment was conducted using 32 NVIDIA H800 GPUs. Each GPU is equipped with 80GB of memory. The training process for our models lasted approximately 12 hours.
>
> In our previous experiments, models with 7B parameters or fewer often performed differently compared to those with 14B, 32B, or even larger parameter counts. To ensure the validity of our experimental conclusions, we opted for models with larger parameter scales whenever possible.
>
> We also experimented with DeepSeek-R1-Distill-Qwen-7B, but encountered significant language inconsistency issues, leading us to abandon this model.
>
> Our primary motivation is to extend RLVR, which has already resulted in substantial improvements in the STEM field, to open-ended tasks.
>
> Througt several ablation studies and tests on different model serials, we find that the key to the effectiveness of the VMR lies in its reliance on specific task types such as multiple-choice or true/false, rather than preference data. This implies that, provided with additional resources and effort, we can indeed the data of convert general open-ended domain into multiple-choice or true/false task, thereby further expanding the applicability of VMR and enhancing its performance in real-world scenarios.
>
> This research is a preliminary step that highlights the need for further exploration of reasoning models in open-ended domains.
>
>
>
>
> **W4:In the bar chart of Figure 1, some numbers overlap with each other.**
>
> Thank you for pointing out the overlap of numbers in the bar chart of Figure 1. We appreciate your attention to detail. We will revise the chart to ensure that the numbers are clearly visible and do not overlap, enhancing readability and comprehension.

---

### Official Review · Reviewer_mkkq · 2025-11-01

**Soundness:** 1
**Presentation:** 1
**Contribution:** 1
**Rating:** 2
**Confidence:** 4

**Summary:**

The paper proposes Verifiable Multiple-Choice Reformulation (VMR), which converts preference pairs (chosen vs. rejected responses) into verifiable binary-choice questions. A rule-based verifier then provides exact rewards (1/0) based on whether the model selects the better response.

**Strengths:**

It is important and anticipated to extend RLVR to open-ended tasks.

**Weaknesses:**

- **The papers lacks methodological novelty and the method is largely a straightforward combination of existing techniques** (e.g., RLVR, GRPO, and preference-based data formatting) without introducing significant conceptual or architectural innovation. While the VMR is a useful engineering trick, it does not constitute a fundamental advance in reinforcement learning or reasoning modeling. The method section is also overly verbose, repeating well-known formulations without sufficient focus on what truly differentiates the proposed pipeline.

- The writing—particularly in the abstract and introduction—lacks focus and fails to clearly articulate the core problem, contribution, and significance. Key claims are buried in lengthy paragraphs, and the narrative does not effectively motivate why extending RLVR to open-ended tasks is non-trivial or why VMR is a principled solution.

- Inadequate formatting and scholarly presentation. The paper suffers from inconsistent or incorrect formatting, including improper citation styles, titles position of tables and position of **REPRODUCIBILITY STATEMENT**.

- Critical implementation details are missing. For instance: The number and type of GPUs used for training are not disclosed; Training time, memory consumption, or computational cost are omitted; Hyperparameter sensitivity or ablation studies (e.g., impact of the 1:1 RM/VMR data mix) are not provided.

- The paper over-relies on LLM-as-a-judge metrics. Most benchmarks (e.g., MT-Bench, AlpacaEval) use automated LLM-based evaluators, which are known to exhibit biases.

- The method is only validated on a single base model (DeepSeek-R1-Distill-Qwen-14B). It remains uncertain whether VMR’s benefits transfer to other architectures, scales, or instruction-tuned models.

**Questions:**

Please see weaknesses.

---

> ### Author Response · Authors · 2025-11-21
> **We sincerely appreciate the time and effort that you have dedicated to evaluating our manuscript.**
>
> **W1:The papers lacks methodological novelty and the method is largely a straightforward combination of existing techniques(e.g., RLVR, GRPO, and preference data formatting) without introducing significant conceptual or architectural innovation.**
>
> Thank you for your valuable feedback. We appreciate your insights and would like to clarify the intent and scope of our work.
> We acknowledge that the primary focus of our paper may have been misunderstood. Our goal was not to introduce novel methods or to advance reinforcement learning algorithms.
> DeepSeek R1 brought about transformative change primarily through the use of RLVR. While RLVR itself may not represent a major innovation in terms of methodology or improvements to reinforcement learning algorithms, its application in the STEM field has been both fundamental and indispensable. We believe that introducing the transformative potential of RLVR into open-ended domains will also prove to be fundamental and indispensable. Such exploration and attempts hold immense potential and should be encouraged and supported.
>
> If our paper is accepted, we plan to present more innovations in the future work. We apologize for any confusion caused by our writing and sincerely thank you for your comments.
>
> **W2:The writing—particularly in the abstract and introduction—lacks focus and fails to clearly articulate the core problem, contribution, and significance. Key claims are buried in lengthy paragraphs, and the narrative does not effectively motivate why extending RLVR to open-ended tasks is non-trivial or why VMR is a principled solution.**
>
> Your feedback is extremely valuable to us, and I have learned a lot from it.
>
> The open-sourcing of DeepSeek R1 brought about a transformative advancement in the STEM field. However, there is rare work focusing on open-ended domains, particularly reasoning models within these domains. These raise a question: can we replicate the significant breakthrough in the STEM fields to open-ended domains?  Our work presents an initial exploration of this possibility.
>
> DeepSeek R1's breakthrough in STEM fields stemmed from RLVR where verifiable problems played a crucial role. Nevertheless, when it comes to general open-ended reasoning with free-form answers, the development of devise rule-based verifiers becomes an impracticable endeavor owing to the high diversity and complexity of natural language. A common approach to overcome this challenge is to train specific LLMs as verifier models, which not only requiring substantial human annotation, but also often achieving unsatisfactory results that cannot meet practical needs.
>
> Against this backdrop, we proposed an alternative, problem-centered solution by converting to verifiable form for open-ended reasoning with free-form answers by task type conversion. Among numerous task types, multiple-choice task stand out for their inherent verifiability, which are not limited by the data domain and have broad applicability. To construct an open-ended training dataset  beloning to multiple-choice task at a low cost, we collect open-source preference data of open-ended domain, which can be easily converted into multiple-choice through simple and clever processing.
>
> Here, it is necessary to emphasize that the key to the effectiveness of the VMR lies in its reliance on specific task types such as multiple-choice, rather than preference data. This implies that, provided with additional resources and effort, we can indeed the data of convert general open-ended domain into multiple-choice task, thereby further expanding the applicability of VMR and enhancing its performance in real-world scenarios.
>
>  We look forward to your further comments and to engaging in more discussions with you.
>
> **W3: Inconsistent or incorrect formatting, including improper citation styles, titles position of tables and position of REPRODUCIBILITY STATEMENT.**
>
> Modified as suggested
>
> **W4:Critical implementation details are missing. For instance: The number and type of GPUs used for training are not disclosed; Training time, memory consumption, or computational cost are omitted; Hyperparameter sensitivity or ablation studies (e.g., impact of the 1:1 RM/VMR data mix) are not provided.**
>
> Each experiment was conducted using 32 NVIDIA H800 GPUs. Each GPU is equipped with 80GB of memory. The training process for our models lasted approximately 12 hours. The ablation studies are provided in Section 3.
>
> **W5:The paper over-relies on LLM-as-a-judge metrics.**
>
> We have conducted experiments in Chinese. The results from human evaluations demonstrate similarly significant improvements. For detailed results, please refer to Section 3.
>
> **W6:The method is only validated on a single base model (DeepSeek-R1-Distill-Qwen-14B). It remains uncertain whether VMR’s benefits transfer to other architectures, scales, or instruction-tuned models.**
>
> We have added experiments based on Qwen3-8B DeepSeek-R1-Distill-Llama-8B Qwen2.5-14B-Instruct in Section 3.

---

### Official Review · Reviewer_Dh1x · 2025-11-03

**Soundness:** 2
**Presentation:** 3
**Contribution:** 1
**Rating:** 2
**Confidence:** 3

**Summary:**

The paper tackles challenges of applying RLVR to open-ended tasks like creative writing. The authors propose a training strategy called Verifiable Multiple-Choice Reformulation (VMR). This method restructures data from open-ended tasks into a multiple-choice format, which makes it possible to verify the answer. The experiments find that the proposed VMR improves the performance or LLMs on open-ended tasks, shoing an average gain of 5.99 points over the baseline.

**Strengths:**

* This paper tries to tackle the LLM training problem that it is not easy to do RL training with open-ended questions. This is an important question that the community tries to solve.
* The empirical results from the proposed method seems good, with a noticeable gain comparing with the baselines.
* The paper is clear that readers can understand most of the concepts introduced easily.

**Weaknesses:**

*  The reward verifies only whether the model selected the pre-labeled preferred response, not that the response is objectively better. For the RM-based subset, line 257, the labels themselves are produced by an automated reward model (URM-LLaMA-3.1-8B). Therefore, the pipeline still inherits RM bias/noise even though the training reward is rule-based. This undercuts the claim that they avoid RM issues (line 063, in figure 2).
* Most reported wins depend on LLM-as-judge (e.g., MTBench, AlpacaEval-2, WildBench, CreativeWriting V3, ArenaHard 2.0, CreativeWriting), which can share stylistic biases with the training signal. There’s no human eval to validate the improvement, making over-optimization to judge preferences a real risk.
* In the experiment, the authors compare with reward model-scored RL baselines. There’s no DPO/KTO (or other RLHF methods) baseline trained on the same pairwise triples, despite those being the most obvious alternatives. Some gains could stem from the extra signal in pairwise data rather than the on-policy RL objective or the VMR prompt itself.
* The A/B candidates come from existing datasets, not from the current policy model, which makes it skeptical if the proposed method can really improve the LLM generation quality
* The proposed method is a form of RLHF, it's just like the actor-critic/PPO loop. The “RM-based dataset” uses open-ended queries whose rewards are assigned by a reward model (URM-LLaMA-3.1-8B). For VMR, each item has a human-labeled chosen vs rejected answer. They convert it to A/B and give a binary reward (1/0) if the policy picks the chosen one (see Figure 3 and Eq (9)). Functionally, that’s RLHF with a degenerate reward model that returns 1 for the preferred option and 0 otherwise. The policy still maximizes expected reward from human preferences via policy gradient.

**Questions:**

* Though the motivation of this paper is to transform open-ended questions into verfiable ones, I wonder what is the necessity of doing so. For training LLMs with RL, is it a good and general enough solution to convert the open-ended questions?

---

> ### Author Response · Authors · 2025-11-21
> **We sincerely appreciate the time and effort that you have dedicated to evaluating our manuscript.**
>
> **W1:For the RM-based subset, the labels themselves are produced by an automated reward model (URM-LLaMA-3.1-8B). Therefore, the pipeline still inherits RM bias/noise. This undercuts the claim that they avoid RM issues.**
>
> We sincerely apologize for any confusion caused. It was not my intention to focus on noise or bias correction. We will delete the figure. To clarify our motivation: in STEM fields, significant improvements have been achieved with RLVR and verifiable problems. We aim to explore whether this substantial enhancement can also be realized in open-ended fields with verifiable problems.
>
> **W2:Most reported wins depend on LLM-as-judge, which can share stylistic biases with the training signal. There’s no human eval.**
>
> Due to language and cultural differences, we conducted several experiments in Chinese, followed by human evaluations to ensure their accuracy. In the Chinese open-domain setting, we utilized multiple-choice questions on subjects such as politics, history, geography, and Chinese exams from middle and high school as verifiable questions. We also employed Chinese queries from the writing and question-answering domain on the Wildchat platform as RM-based queries. By integrating these verifiable problems into our training process, we achieved performance that significantly surpassed that of BaselineV1, which relied solely on RM-based queries for training. Detailed results from the human evaluations can be found in Section 3.
>
> **W3:In the experiment, the authors compare with reward model-scored RL baselines. There’s no DPO/KTO (or other RLHF methods) baseline trained on the same pairwise triples, despite those being the most obvious alternatives. Some gains could stem from the extra signal in pairwise data rather than the on-policy RL objective or the VMR prompt itself.**
>
> Thank you for your valuable feedback. We have included additional experiments on DPO in Section 3. The results of these experiments demonstrate that the improvements in VMR are not attributable to the query-chosen-rejected pairwise data. We appreciate your insights and hope that the DPO experiment provides clarity on the source of VMR’s performance enhancements. Thank you once again for your guidance and support.
>
>
> **W4:The A/B candidates come from existing datasets, not from the current policy model, which makes it skeptical if the proposed method can really improve the LLM generation quality**
>
> Thank you for your insightful comments. We would like to clarify that while the A/B candidates are not derived from the current policy model, this does not impact the effectiveness of our approach.
> During the training process, the model does not explicitly increase the probability of chosen responses or decrease the probability of rejected ones.
> Instead, it learns from the comparison of A/B candidates to identify which patterns are more effective. For example, responses that are more organized or more eloquent tend to perform better. These patterns are generalizable and contribute to the robustness of our model.
> We appreciate your feedback and hope this explanation provides a clearer understanding of our method’s strengths.
>
>
> **W5: They convert it to A/B and give a binary reward (1/0) if the policy picks the chosen one (see Figure 3 and Eq (9)). Functionally, that’s RLHF with a degenerate reward model that returns 1 for the preferred option and 0 otherwise. The policy still maximizes expected reward from human preferences via policy gradient.**
>
> Apologies for the lack of clarity in my writing, which failed to effectively convey why extending RLVR to open-ended tasks is non-trivial or why VMR is a principled solution.
>
> Our work investigates whether incorporating verifiable problems can enhance performance in open-ended domains. This research represents a preliminary step in extending Reinforcement Learning with Verifiable Rewards (RLVR), which has already led to significant improvements in the STEM field, to open-ended tasks.
>
> Through extensive ablation studies and experiments on different serial models, we can conclude that the effectiveness of the VMR primarily hinges on its reliance on specific task types such as multiple-choice questions, rather than preference data. This suggests that, with additional resources and effort, it is possible to transform data from a general open-ended domain into multiple-choice or verifiable problems. This transformation can further broaden the applicability of VMR and enhance its performance in real-world scenarios, similar to the breakthrough brought by RLVR in the STEM field.
>
> **Q1:Though the motivation of this paper is to transform open-ended questions into verfiable ones, I wonder what is the necessity of doing so. For training LLMs with RL, is it a good and general enough solution to convert the open-ended questions?**
>
> Thank you for your question, which allows us to better clarify our motivation.  Please refer to the response to Weakness 5.
>
> We look forward to further discussion.

---

### Official Review · Reviewer_Hi7t · 2025-11-03

**Soundness:** 3
**Presentation:** 3
**Contribution:** 3
**Rating:** 6
**Confidence:** 2

**Summary:**

This paper extends Reinforcement Learning with Verifiable Rewards (RLVR) from STEM domains (mathematics, programming) to open-ended tasks lacking ground-truth solutions (creative writing, instruction following). The key innovation is Verifiable Multiple-Choice Reformulation (VMR), which restructures preference data (chosen/rejected response pairs) into multiple-choice questions that can be verified using rule-based functions. For each query, the model is asked to choose between two randomly-ordered responses, and receives binary reward based on selecting the better one.

**Strengths:**

- Novel extension of RLVR to open-ended domains where standard answers don't exist
- Sound mathematical formulation connecting VMR to standard RLVR framework
- Clear problem motivation explaining RLVR's limitation in open-ended domains
- Addresses important gap: extending RLVR beyond STEM domains

**Weaknesses:**

- The connection between multiple-choice discrimination and open-ended generation is assumed but not justified
- Only one base model tested (DeepSeek-R1-Distill-14B); crucial to test on other models
- Dependency on high-quality preference data limits applicability
- Heavy reliance on LLM-as-judge metrics which have known biases

**Questions:**

- How does VMR perform on models without built-in reasoning capabilities ?
- Can you provide error bars or significance tests for the improvements?
- The reasoning density improvement is quite small. Is this statistically significant?
- How does the method perform when preference annotations disagree or are noisy?

---

> ### Author Response · Authors · 2025-11-20
> **We sincerely appreciate the time and effort that you have dedicated to evaluating our manuscript.**
>
> **W1:The connection between multiple-choice discrimination and open-ended generation is assumed but not justified**
>
> Most RLVR methods are confined to stem domain. The primary reason for this limitation lies in the fact that existing RLVR approaches heavily depend on domain-specific verifiers, such as handcrafted rules, to derive rewards. Nevertheless, when it comes to general open-ended reasoning with free-form answers, the development of devise rule-based verifiers becomes an impracticable endeavor owing to the high diversity and complexity of natural language. A common approach to overcome this challenge is to train specific LLMs as verifier models, which not only requiring substantial human annotation, but also often achieving unsatisfactory results that cannot meet practical needs.
>
> Against this backdrop, we proposed an alternative, problem-centered solution -- converting to verifiable form for open-ended reasoning with free-form answers by task type conversion. Among numerous task types, multiple-choice task stand out for their inherent verifiability, which are not limited by the data domain and have broad applicability. To construct an open-ended training dataset  beloning to multiple-choice task at a low cost, we collect open-source preference data of open-ended domain, which can be easily converted into multiple-choice task through simple and clever processing.
>
> Here, it is necessary to emphasize that the key to the effectiveness of the VMR lies in its reliance on specific task types such as multiple-choice, rather than preference data. This implies that, provided with additional resources and effort, we can indeed the data of convert general open-ended domain into multiple-choice task, thereby further expanding the applicability of VMR and enhancing its performance in real-world scenarios.
>
>
> **W2:Only one base model tested (DeepSeek-R1-Distill-14B); crucial to test on other models**
>
> Due to time and resource constraints, experiments were initially conducted only based on  the DeepSeek-R1-Distill-14B model prior to the paper submission deadline. However, subsequent experiments based on the **Qwen3-8B** **DeepSeek-R1-Distill-Llama-8B** model were completed after the initial submission. The performance improvement observed in these experiments further validates the effectiveness of the VRM algorithm, which will be add in Section 3.
>
> **W3:Dependency on high-quality preference data limits applicability**
>
> We acknowledge the concern regarding the reliance on high-quality preference data. In our work, we have explored the integration of verifiable problems into reasoning models within open-ended domains. This exploration aims to determine whether such an approach can yield improvements similar to those achieved by RLVR in STEM fields. Importantly, our method is not limited to multiple-choice questions derived from preference data; it can be applied to any form of verifiable problem, thereby broadening its applicability.
>
> **W4:Heavy reliance on LLM-as-judge metrics which have known biases**
>
> Due to language and cultural differences, we conducted several experiments in Chinese, followed by human evaluations to ensure their accuracy. In the Chinese open-domain setting, we utilized multiple-choice questions on subjects such as politics, history, geography, and Chinese exams from middle and high school as verifiable questions. We also employed Chinese queries from the writing and QA domain on the Wildchat platform as RM-based queries. By integrating these verifiable problems into our training process, we achieved performance that significantly surpassed that of BaselineV1, which relied solely on RM-based queries for training. Detailed results from the human evaluations can be found in Section 3.
>
>
> **Q1:How does VMR perform on models without built-in reasoning capabilities ?**
>
> We conducted experiments on the Qwen2.5-14B-Instruct model (non-reasoning model), and the experimental results are detailed in section 3.
>
>
> **Q2:Can you provide error bars or significance tests for the improvements?**
>
> We are committed to enhancing the statistical validation of our results and will be supplementing our current data with error bars in the following week.
>
>
> **Q3:The reasoning density improvement is quite small. Is this statistically significant?**
>
>
> The calculation method for reasoning density is reasoning steps/char length, which is why the reasoning density is represented by such a small number. An increase from 0.021 (**47.6 token per reasoning step**) to 0.026 (**38.5 token per reasoning step**) represents a **23% growth**, which is a statistically significant improvement.
>
> **Q4:How does the method perform when preference annotations disagree or are noisy?**
>
> The multiple-choice questions used in the article are not all manually labeled. Some of them are labeled automatically, which inherently introduces some noise and disagreement.
>
> **Our:**
>
> Once again, we are grateful for the insightful comments provided by you.

---

### Note · Authors · 2026-01-09

**Comment:**

Dear Conference Organizers,

I am writing to request the withdrawal of our paper titled “Extending RLVR to Open-Ended Tasks via Verifiable Multiple-Choice Reformulation” with the paper ID 18388, submitted to ICLR. The reason for this withdrawal is that we have identified some issues.

We greatly appreciate the time and effort that you and the reviewers have dedicated to evaluating our work. We hope to resubmit once we have refined our research.

Thank you for your understanding and support.

Sincerely,

Mengyu Zhang

**Withdrawal Confirmation:**

I have read and agree with the venue's withdrawal policy on behalf of myself and my co-authors.